# The Promising Role of Non-Coding RNAs as Biomarkers and Therapeutic Targets for Leukemia

**DOI:** 10.3390/genes14010131

**Published:** 2023-01-03

**Authors:** Mohammad H. Ghazimoradi, Naeim Karimpour-Fard, Sadegh Babashah

**Affiliations:** 1Department of Molecular Genetics, Faculty of Biological Sciences, Tarbiat Modares University, Tehran 1411713116, Iran; 2Department of Pharmacoeconomics and Pharmaceutical Administration, Faculty of Pharmacy, Tehran University of Medical Sciences, Tehran 1417614411, Iran

**Keywords:** leukemia, biomarkers, non-coding RNAs, miRNA, lncRNA, circRNA, piRNA

## Abstract

Early-stage leukemia identification is crucial for effective disease management and leads to an improvement in the survival of leukemia patients. Approaches based on cutting-edge biomarkers with excellent accuracy in body liquids provide patients with the possibility of early diagnosis with high sensitivity and specificity. Non-coding RNAs have recently received a great deal of interest as possible biomarkers in leukemia due to their participation in crucial oncogenic processes such as proliferation, differentiation, invasion, apoptosis, and their availability in body fluids. Recent studies have revealed a strong correlation between leukemia and the deregulated non-coding RNAs. On this basis, these RNAs are also great therapeutic targets. Based on these advantages, we tried to review the role of non-coding RNAs in leukemia. Here, the significance of several non-coding RNA types in leukemia is highlighted, and their potential roles as diagnostic, prognostic, and therapeutic targets are covered.

## 1. Introduction

A complex interaction of regulatory molecules controls the balance between self-renewal, proliferation, quiescence, and cell identity of hematopoietic stem cell homeostasis and lifelong blood production [1,2]. If this vital equilibrium is disrupted, myelodysplasia/ myeloproliferative neoplasms, or the worst scenario, leukemia, may occur [3,4]. In addition to the breakdown of organs, the growth of leukemic cells drives the collapse of hematopoietic cells and the loss of their activities. This produces severe symptoms, including thrombocytopenia, anemia, and immunodeficiency [5]. Acute or chronic progression patterns and afflicted lineages are used to classify leukemia (lymphoid (Precursor of T and B cells) or myeloid (Precursor of platelets, erythrocytes, and monocytes). Acute lymphoblastic leukemia (ALL), chronic lymphocytic leukemia (CLL, the most widespread type), acute myeloid leukemia (AML), and chronic myeloid leukemia (CML) are the four primary types. However, the relationship of these four types to the lineage of cells also could be taken into account. In ALL, T cell or B cell lineage could be seen under the name of T-ALL or B-ALL. On the other hand, AML could be divided into monoblast, erythroblast, and megakaryoblast AMLs which could be determined by lineage tracing and surface markers, in addition to cytogenetics abnormalities [6]. Although the other subtypes are more prevalent in adults, ALL is the most pervasive kind of leukemia in children globally [7]. The creation of functionally healthy cells is hampered in leukemias by the aberrant proliferation of leukemic cells. Leukemia patients get anemia, impairing their ability to fight infections and blood clotting issues [8]. The majority of leukemia causes and its subtypes remain unknown, partly because there are many different abnormalities and risk factors. While most of the research investigated protein-coding genes as leukemia’s driving factors, 97% of the human genome has been formed by non-coding RNAs (ncRNA) (Figure 1) [9]. 

Based on the length, ncRNAs can be divided into three categories: short (19–31 nucleotides), mid–200 nucleotides, and long (>200 nucleotides) [10]. Another way to characterize these RNAs is based on their function (Figure 2). While some RNAs control cells as a regulatory element, others have structural duties. MicroRNAs (miRNAs), which are short non-coding RNAs (22–25 nucleotides), and long non-coding RNAs (lncRNAs), which comprise the most significant class of ncRNAs, are among the most studied ncRNAs in cancers [11]. LncRNAs may control gene expression through their interaction domains with DNA, mRNAs, miRNAs, and proteins. In contrast, miRNAs may mediate post-transcriptional gene regulation by translational repression or mRNA degradation via miRNA-mRNA complementarity [12,13]. Circular RNA, a nonlinear non-coding RNA, has a significant role in cancer via sponging miRNA, regulation of genes, and producing polypeptides with some functions [14]. Deregulated expression of these elements has been seen in many malignancies. This deregulation could alter the expression or function of downstream targets, leading to diseases including leukemia [15,16]. Deregulated circRNAs or lncRNAs could also change miRNA availability via sponging them, leading to the deregulation of downstream targets of miRNA, including but not limited to oncogenes and tumor suppressors [15,17]. rRNA, tRNAs, snRNAs, and snoRNAs are the most significant RNA in structural RNA groups with roles in the ribosome, spliceosome, and other complexes [18,19]. Deregulation of these RNAs could be a potential therapeutic target, and abnormal expression of these factors could indicate hemostasis deregulation of normal cells via affecting oncogene and tumor suppressors and could be a potential biomarker with high specificity in various stages of cancers (Figure 2). Fortunately, these molecules are available in multiple body parts and are tissue-specific. In addition, a substantial number of these elements give us several options to consider [20]. All of the mentioned factors and advancements in precise and high-throughput detection methods make these RNAs ideal as biomarkers and therapeutic targets, especially in leukemia. Here, we try to summarize the current knowledge on the patterns of ncRNA expression in various leukemias, the contribution of ncRNA to leukemia carcinogenesis, and their functions. Understanding the crucial function and pattern of non-coding RNAs may enhance our understanding of their biological processes and, in turn, lead to the discovery of potential therapeutic targets, creating many opportunities for treatment and diagnosis. Here, we try to summarize the roles of ncRNAs in leukemia while almost all of them could be considered biomarkers (Table 1).

## 2. Regulatory Non-Coding RNAs as Biomarkers and Therapeutic Targets in Leukemia

### 2.1. MicroRNAs

MicroRNAs (miRNAs) are between 19 and 20 nt in length. The ribonuclease Dicer generates miRNAs from their precursors, which have a unique hairpin secondary structure. MiRNAs were initially found in Caenorhabditis elegans and are present in most eukaryotes, including humans. According to some estimates, miRNAs make up from one to five percent of the human genome [84,85]. So far, several miRNA compounds have been identified. Although less known about the functions and targets of miRNA molecules, it is obvious that they control several biological processes and gene expression regulation [86]. Any deregulation of these molecules causes hemostasis disruption, resulting in diseases such as cancer. In malignancies, particularly leukemia, oncogenic miRNAs are overexpressed, whereas tumor-suppressing miRNAs are frequently downregulated [87]. It has been demonstrated that the equilibrium between miR-194-5p and its target BCL2-associated transcription factor 1 (BCLAF1) is compromised in AML cases [21]. By specifically targeting IGF-1R, miR-628 prevents acute myeloid leukemia cells from proliferating [22]. Furthermore, it was shown that miR-10a-5p is overexpressed in AML and controlled via differentiation [88]. miR-143, via reduction of ATG7- and ATG2B-dependent autophagy, makes acute myeloid leukemia cells more susceptible to cytarabine [23]. miR-96 has been associated with leukemic burden, and its expression was downregulated in AML [89]. miR-155 is increased in hematopoietic stem cells from AML specimens with FLT3-ITD and nucleophosmin (NPM1) gene mutations [90]. Similar to this, it has been discovered that increased miR-155 expression in murine lymphocyte progenitors results in polyclonal lymphocytosis and raises the risk of developing high-grade lymphocytic leukemia [91]. There have also been instances of people with myeloproliferative disorders having overexpression of the miR-155 gene, which causes an increase in granulocyte–monocyte cells. The miR-155 targets are SHIP1 and CEBPB, both crucial for granulopoiesis and aberrantly expressed in AML patients. It is interesting to note that miR-126 downregulation controls both routes [24]. MiR-133 downregulates Ecotropic viral integration site 1 (Evi1) in leukemic cells, increasing treatment sensitivity, according to Yamamoto et al. As a result, it is feasible that miR-133 might be used to treat leukemias that overexpress Evi1 [26]. miR-223 was found to be significantly expressed in AML cell lines and to impair cell motility and proliferation while increasing apoptosis [27]. Research has shown that ectopic miR-223 overexpression inhibits cancer by controlling the G1/S cell cycle [27,28,29]. Lin’s study found that AML patients had a lower miR-370 expression, which likely had a role in the disease’s development. Additionally, it has been proposed that the expression of miR-370 might act as a non-invasive biomarker for children with AML [92]. Let-7a, miR-99b, miR-146a, miR-150, miR-155, miR-191, and miR-1246 were among the miRNAs Hornick et al. identified as being elevated in AML exosomes from the serum of mice model. These exosomal miRNAs might be administrated as a detection method for early AML. miR-26a-5p and miR-101-3p were also markedly elevated in AML, whereas miR-23b-5p, miR-339-3p, and miR-425-5p were reduced in exosomes [93]. Another study found high levels of miR-150 and miR-155 in exosomes isolated from in vitro and in vitro in AML [94]. miR-216b may be a helpful biomarker for disease recurrence in AML and is a prognostic factor for cytogenetically normal acute myeloid leukemia (CN-AML) independently [95]. In CN-AML, those who expressed high levels of miR-362-5p had shorter overall survival and were mostly elderly [96]. Diaz-Beya et al. discovered that compared to AML patients with low expression of miR-3151, those with high expression had shorter survival time, lower rates of CR, and greater cumulative incidences of recurrence [97]. miR-328 levels in AML patients may be used as a predictive biomarker in individuals who express little miR-34a [98]. As a tumor suppressor in AML and a standalone predictor of disease prognosis, miR-135a has been found [99]. Patients with elevated expression of miR-146a and miR-3667 have better outcomes than patients with lower expression levels [100]. In contrast, the downregulation of miR-122, miR-192, miR-193b-3p, miR-204, and miR-217, as well as miR-340, is a weak predictor of AML [30]. In the leukemic cell lines HL60, NB4, and K562, Liu discovered upregulating miR-181a results in more cell proliferation and enhanced cell cycling [31]. After miR-128 transfection, HL60 cell lines showed an increase in apoptosis, genome instability, and drug sensitivity, although the molecular mechanism is yet unclear [101]. According to Volinia, MiR-128 was shown to be overexpressed and increased in numerous malignancies, despite its expression being downregulated in AML cells with NPM1 mutations [102]. Juvenile acute promyelocytic leukemia contained greater levels of miRNA-125b than other acute myelogenous leukemia (AML) subtypes, and exogenous expression of this gene in AML cells resulted in DOX resistance [103]. ALL patients with multidrug resistance are reported to have significantly reduced levels of miR-326 expression compared to the non-resistance group [104]. According to a recent study, the miR-125b-2 cluster expression was elevated in leukemia patients with the ETV6-RUNX1+ fusion gene. In addition, overexpression of the miR-125b-2 cluster increased cell survival and lowered apoptosis in REH ETV6-RUNX1+ cells, indicating that this miRNA may have therapeutic potential for treating pediatric ALL [32]. miR-221 and miR-222 significantly upregulated in the high-risk subgroup of Early T-cell precursor ALL (ETP-ALL) patients compared to non-ETP-ALL patients. Furthermore, it has been hypothesized that miR-222, which suppresses ETS1 expression, may partly be responsible for the myeloid character of ETP-ALL [33]. miR-19, the cluster component with the highest expression in T-cell acute lymphoblastic leukemia (T-ALL) in a mouse model of Notch1-induced T-ALL, increases lymphocyte survival and induces leukemogenesis [105]. By targeting Forkhead box O1, miR-223 increases cell apoptosis while reducing cell proliferation, migration, and invasion in ALL [106]. The MiR-223 has been classified as having a pronounced upregulation in T-ALL. It was shown that, at least in part, it contributed to the induction of Notch1-driven leukemia by controlling the E3 ligase FBXW7 [34]. miR-142-3p was identified as a crucial microRNA of hematopoietic stem cells [107]. It is substantially expressed in ALL samples, especially in T-ALL children with poor prognoses, as compared to donor T-cells in the healthy group [108]. It has been demonstrated that MiR-142-3p causes glucocorticoid (GC) therapy resistance in T-ALL by promoting leukemic cell proliferation with significant roles in proliferation and chemoresistance features [35]. miR-196b is over-expressed in T-ALL patient samples compared to B-cell acute lymphoblastic leukemia (B-ALL) patient samples. miR-196b also is co-expressed with genes of the HOXA cluster in T-ALL. When overexpressed in mouse bone marrow cells, this miRNA increases those cells’ proliferative capability and survival [109,110]. By targeting PIK3CD, the tumor suppressor MiR-26b blocks the PI3K/AKT pathway [36]. PTEN stimulates the expression of miR-26b through the regulation of isoforms of Ikaros [37]. miR-181a contributes to Notch oncogenic signals in T-ALL by lowering Notch negative feedback and enhancing pre-TCR signals [111]. LeukmiR, a notable online database, is used to identify and classify microRNAs that have a potential or established influence on ALL [112]. Another microRNA linked to T-ALL is miR-30a, which is transcriptionally inhibited by MYC and is an inhibitor of NOTCH1 and NOTCH2 [38]. miR-146b-5p is a tumor suppressor inhibited by TAL1 and is engaged in T-cells [113]. MiRNAs have also contributed to chronic myeloid leukemia and tumor drug resistance. According to recent studies, the miR-221/STAT5 axis played a significant role in modulating how responsive CML cells were to imatinib (a tyrosine kinase inhibitor) [39]. miR-497/195 via CDKN2A/B inhibits CLL tumor growth [40]. Another research asserts that the lncRNA MALAT1/miR-328 axis promotes imatinib resistance and cell proliferation, opening up new research directions for MALAT1 as a possible therapeutic target in CML [41]. miR-214 was also connected to imatinib resistance in CML through ABCB1 regulation [42]. CML Transition from Chronic Phase (CP) to Blast Crisis (BC) is promoted by miR-142 Deficiency [114]. miR-150 is down-regulated, which makes it a strong candidate for the early identification of CML, according to several studies that have repeatedly demonstrated this. miR-150 is linked to a poor prognosis and a more advanced stage of CML, according to several studies [115,116]. Using patient samples with varying stages of CML, it has been shown that miR-150 serves as a biomarker and therapeutic target. miR-150 has reduced expression in both CP and BP stages. Importantly, miR-150 levels have not been recovered in patients who developed resistance to imatinib therapy [117]. The miR-17/92 cluster has a lot of promise for use as a diagnostic marker [118]. miR-17, miR-18a, miR-19a, miR-19b-1, miR-20a, and miR-92a-1 are the six miRNAs that make up this cluster. It has been established that human B-cell lymphomas, such as diffuse large B-cell lymphomas, Hodgkin lymphomas, and particular kinds of Burkitt lymphomas, accumulate miR-155 and BIC in K562 cells when the miR-17/92 cluster is overexpressed [119]. It has been found that the miR-155 in CLL changes SHIP1 expression in favor of cancer [25]. The miR-17/92 cluster is a potential marker of CLL progression and is overexpressed in CD34+ cells of CP but not BP [120]. MiR-10a level was reduced in 71% of CML cases, highlighting the clinical significance of this biomarker for diagnosis [121]. miR-10a targets USF1, a cell proliferative transcription factor that has been extensively investigated and is elevated in 60% of CP CML cases [43]. miR-203 has been suggested as a key RNA to manage CML. It has been shown that miR-203 knockdown reduces BCR-ABL expression, which in turn slows the rate of proliferation [44]. Because MiR-29a/b is constantly reduced in CML and because its expression corresponds with imatinib resistance, it is a significant biomarker for therapy response [122,123]. It is likely that miR-29b targets BCR-ABL since it is detected at minimal levels in patient samples. It was found that miR-29b targets Abl-1, proved by luciferase assay. Overexpression of miR-29b reduces ABL-1 level and induces G1 phase cell arrest via activation of p21 and p27. In K562, miR-29b promotes apoptosis and Caspase 3 activity [45]. It has been demonstrated that miR-320 targets ABL with involvement in tumor resistance phenotype and inhibits the translation of the ABL protein [46]. miR-181a and miR-181b are overexpressed in CLL and make the cells more vulnerable to fludarabine-mediated cell death [47]. Similar to this, miR-181b restoration lowers MCL-1 and HMGB1 expression, increasing sensitivity to doxorubicin (DOX) and cytarabine (ara-C) [48]. Overexpression of miR-125b resulted in resistance to daunorubicin and reduced apoptosis by inhibiting GRK2 and PUMA [49]. In over 66% of CLL cases, miR-15a/16-1 expression was reported to be downregulated [124]. A point mutation in the miR-15a/16-1 decreased the expression of miR-16-1 in NZB lymph and upregulated Bcl-2 levels [125,126]. CLLs overexpress miR-29 compared to normal B cells, in contrast with miR-181b. However, in aggressive CLLs, miR-29 and miR-181b show greater expression [127]. 

### 2.2. Long Non-Coding RNAs Role in Leukemia

NcRNAs with more than 200 nucleotides are known as long non-coding RNAs. Intergenic, intron, sense, and antisense lncRNAs are the several types of lncRNAs [128]. In cells, different lncRNAs may function as a guiding RNA that binds to RNA binding and control the expression of the gene, a scaffold, acting as a hub for other molecules, a miRNA sponge, and a signal molecule with differential expression pattern [129]. The cell cycle and tumor invasion are two biological processes in which lncRNAs are engaged, some of which serve as oncogenes and others as tumor suppressors [130]. Here we try to summarize the latest advances and contributions of lncRNAs in leukemia. One of the most impact full lncRNA proposed in AML is ZNF571-AS1. This lncRNA is a possible modulator of JAK, STAT 5A, and KIT based on correlation analysis [131]. Another long non-coding with useful applications is LNC00899. It is interesting to note that The Lnc00899 serum levels could be recognized as a prognosis and diagnosis marker of AML [132]. The tumor suppressor MEG3 has been linked to a considerably worse overall survival rate in AML. Numerous human malignancies are associated with this gene [133,134]. MEG3 hypermethylation was discovered to be present in 47.6% of AML cases, which may be related to a much worse overall survival rate in these patients, according to an analysis of the aberrant promoter methylation of MEG3 in 42 AML patients [135,136,137]. In pre-B ALL cases, BALR-1, BRL-6, and LINC0098 were upregulated, and all three of these genes were associated with cytogenetic abnormalities and patient survival rates in B-ALL [138]. LncRNA RP11-137H2.4 significantly influenced apoptosis, proliferation, and cell migration; silencing it is sufficient to restore an NR3C1-independent cellular response to glucocorticoid, resulting in glucocorticoid-induced apoptosis in resistant B-ALL cells [50]. Both T- and B-ALL patients had downregulation of linc-PINT [139]. In T-ALL cells, NOTCH1 wild type and mutant have differential expression of LUNAR1 and lnc-FAM120AOS-1 [140]. MALAT1 was shown to be elevated in K562 cells, and its knockdown slowed cell growth and halted the cell cycle via targeting miR-328 [141]. Huang et al. showed that acute monocytic leukemia patients (AML-M5) had upregulated MALAT1 expression compared to the healthy group [142]. Wang et al. demonstrated that lncRNA CRNDE is increased in AML patients, particularly in acute myelomonocytic leukemia (AML-M4) and AML-M5. It has been shown lncRNA CRNDE promotes cell cycle, proliferation, and suppression of apoptosis in the U937 cell line [51]. LncRNA PVT1 has been linked to carcinogenesis, tumor stage, and poor prognosis in a variety of cancer types [143]. PVT1 is upregulated in ALL and AML patients’ bone marrow and peripheral blood mononuclear cells compared to healthy people [144]. PVT1 knockdown in Jurkat cells increased the rate of apoptosis, caused a G0/G1 cell cycle arrest, decreased proliferation, and decreased the stability of the c-Myc protein [52]. lncRNA TUG1 was shown by Wang et al. to be increased and demonstrated that lncRNA TUG1 lowered apoptosis and increased cell proliferation in AML cells in vitro, indicating that it may be involved in the AML via modulation of cell proliferation [53]. By serving as a miR155-competing endogenous RNA, CCAT1 has been shown to enhance cell proliferation and block myeloid cell differentiation [54]. It has been demonstrated that lncRNA NR-104098 demotes AML proliferation and promotes differentiation by interacting with E2F1 to repress EZH2 transcription [55]. LncRNA CCD26 was shown to be elevated in patients with ALL and AML [145]. Patients with AML also had greater levels of LINC00265 expression in their blood and bone marrow. Furthermore, this lncRNA may be a predictor of prognosis for AML, and its knockdown reduced the proliferation, migration, and invasion of AML cell lines while increasing apoptosis via the PI3K-Akt pathway [56]. NEAT1 expression is downregulated in CML, and its knockdown increased imatinib-induced apoptosis [146]. NEAT1 downregulation was seen in primary AML. Additionally, compared to peripheral white blood cells, the expression of this lncRNA was reduced in the HL-60, Jurkat, and K562, and its overexpression improved the cytotoxic compound-induced multidrug resistance phenotype [57]. A crucial antagonistic regulator of Bcr-Abl-induced carcinogenesis is lncRNA-IUR1. Cells from individuals with CML that were Bcr-Abl positive only very weakly produced LncRNA-IUR1 [147]. The expression of HOTAIR was connected with clinical-pathological prognostic classification in AML; it has been proposed as a possible marker of prognosis and prospective therapeutic target of AML and CML [58]. Furthermore, in AML, silencing of HOTAIR reduced the number of colony-forming cells, triggered apoptosis, and hindered cell proliferation. In individuals with high MRP1 expression and K562-imatinib-resistant cells, HOTAIR appears to have a role in acquired resistance to imatinib in CML. Conversely, when HOTAIR was knocked down, MRP1 expression was decreased, increasing sensitivity to imatinib therapy [59]. Additionally identified as a CML diagnostic biomarker is lncRNA CCAT2. This non-coding RNA may also forecast a patient’s response to imatinib [148]. LncRNA SGNH5 was elevated in K562-imatinib-resistant cells versus normal K562 cells and CML patients. It also has been shown that imatinib resistance in K562 cells was increased by SNHG5 overexpression and that imatinib resistance could be decreased in K562-imatinib-resistant cells by SNHG5 knockdown [60]. Other research confirmed that the expression of the lncRNA SNHG5 was greater in AML and ALL patients compared to the control groups [149]. Through chromatin interaction, LncRNA Hmrhl has also been demonstrated to have functional roles in controlling the expression of genes linked to malignancy in chronic myelogenous leukemia [61].

### 2.3. Circular RNAs and Circ/Mir Axis

CircRNAs are nonlinear ncRNAs that have been shown to have strong potential as prognostic, diagnostic, and predictive biomarkers. This is especially true given that these ncRNAs are easily detectable in liquid biopsies such as plasma, saliva, and urine. However, the function of circRNAs in leukemia needs a lot of research [150]. Here is a summary of recent progress in this area. According to recent research, AML patients express more circ-DLEU2 than healthy controls. Increased cell proliferation in vitro, slowed apoptosis, and enhanced tumor development in vivo were all associated with higher circ-DLEU2 expression [62]. Through miR-134-5p and SSBP2, circ-0004277 prevents AML from progressing. The direct target of circ-0004277 is MiR-134-5p [63]. CircSPI1 inhibits SPI1 and interacts with miR-1307-3p, miR-382-5p, and miR-767-5p to promote proliferation and prevent apoptosis, functioning as an oncogene in AML [64]. FLT3 WT cells did not respond to CircMYBL2’s effects on apoptosis, proliferation, or cell-cycle progression in FLT3-ITD leukemic cells. Additionally, circMYBL2-KD decreased FLT3 kinase expression, which modulated downstream signaling, such as decreased phosphorylation of STAT5. Additionally, by making it easier for PTBP1 to bind to FLT3 mRNA, circMYBL2 improved FLT3 kinase translation. The reversal of these effects on the expression levels of MCL1 and p27/Kip1 was likewise connected to CircMYBL2-KD. CircumMYBL2-KD was shown to decrease FLT3-ITD AML progression and is related to the survival of mice with quizartinib susceptible and resistant AML. On this basis, circMYBL2 may act as a mediator (or at least modulator) of driver mutations in AML biology [65,66,67]. Pediatric AML patients have considerably higher levels of Circ-004136 expression. Circ-004136 controls the growth of AML cells by sponging miR-142 and miR-29a. Another investigation revealed that circRNF13 was upregulated in AML patients. By functioning as a sponge for miR-1224-5p, circRNF13 down-regulation decreased proliferation, induced cell cycle arrest, enhanced apoptosis, and reduced migration and invasion of AML cells. Caspase 3/7 activation, decreased Tenascin-C expression, and c-MYC regulation contributed to these outcomes [68,69]. Leukemia cells have been found to have downregulated levels of circ-0121582 and GSK3beta. In the cytoplasm, circ-0121582 serves as a miR-224 sponge, while in the nucleus, it attaches to the GSK3beta promoter to enlist the DNA demethylase TET1. The Wnt/β-catenin signaling pathway is inhibited due to the increased GSK3beta expression, which promotes the growth of leukemia cells. Patients with AML had higher levels of the gene has-circ-100290 [70]. Hsa-circ100290-KD decreased cell proliferation and accelerated apoptosis via miR-203, leading to the regulation of cyclin D1, CDK4, BCL-2, and cleaved Caspase-3 expression. miR-203 also controlled the expression of RAB10, a small GTPase belonging to the RAS superfamily [71]. Hsa-circ-0079480 increased HDGF expression via sponging miR-655-3p in AML cell lines. In the same research, low miR-655-3p expression and high HDGF expression are linked to worse survival in AML cases. By targeting its neighbor genes c-Myc and Bcl-2, circPVT1 silencing caused cell cycle arrest and death in ALL cells. AML patients had greater levels of hsa-circ-0079480 expression than those with idiopathic thrombocytopenia. AML cells with hsa-circ-0079480-KD had decreased viability and had more apoptosis [72]. Hsa-circ-0002483 was upregulated in AML cases. By reducing BCL-2 and elevating BAX and C-caspase-3, hsa-circ-0002483-KD reduced cell proliferation, cell cycle arrest (G0/G1), and enhanced apoptosis. By sponging miR-758-3p, it also increased MYC expression [73]. Both hsa-circ-0001857 and hsa-circ-0012152 could clearly distinguish between ALL and AML [151]. TKI resistance was linked to CircBA1, which raised ABL1 and BCR-ABL1 protein expression levels. When circBA1 is overexpressed in leukemia cells, it may promote cell growth and medication resistance [74]. A circRNA generated from CRKL called circCRKL controls BCR-ABL by sponging miR-877-5p to enhance the proliferation of CML cells [75]. In the PBMCs of CML patients, hsa-circ-0058493 is overexpressed, and an increased level of circ-0058493 was linked to imatinib’s subpar clinical effectiveness. Significantly reducing the expression of circ-0058493 prevented CML cells from becoming imatinib-resistant. In CML cells that had circ-0058493 downregulated, miR-548b-3p was overexpressed. Circ-0058493 may exert its regulatory role by functioning as a “sponge” for miR-548b-3p, according to a bioinformatic study [76]. Additionally, hsa-circ-0058493 was markedly concentrated in the exosomes produced by CML cells resistant to imatinib [76,152]. Circa-CBFB may develop into a viable therapeutic target for CLL therapy, given the relationship between the signaling cascade and CLL development [153]. Wu et al. have shown a potential regulatory mechanism that circ-0132266 is involved in the downregulation of miR-337-3p [77]. By targeting the miR-1283/WEE1 Axis, circZNF91 promotes the Malignancy of CLL Cells. WEE1 expression was increased by the inhibitor of miR-1283, whereas it was decreased by silencing circ-TTBK2. CircZNF91’s influences on cell proliferation, apoptosis, and cell cycle regulation were mediated through the miR-1283/WEE1 axis [78]. Circular RNA from the Mitochondrial Genome, mc-COX2, Is an Oncogene in Chronic Lymphocytic Leukemia. In CLL cells, mc-COX2 silencing boosted the anti-tumor effects of medicines used in conjunction. It has been shown that the development and prognosis of CLL are related to mc-COX2, which is generated from CLL cells and supplied by exosomes [154]. 

### 2.4. PiRNA Role in Leukemia

Piwi-interacting RNAs (piRNAs) are ncRNAs that were just recently discovered. The human genome currently contains over 20,000 piRNA genes. These RNAs were first identified as essential regulators for transposon silencing and germline maintenance [155]. Numerous studies have demonstrated that the transcriptional activation of retrotransposon elements induced by mutations of vital proteins found in piRNA pathways increases DNA damage in germ cells. Intriguingly, recent research has shown that a subpopulation of piRNAs may have a significant impact on other epigenetic mechanisms, including control of heterochromatin formation, histone modifications, post-transcriptional modifications, and polycomb group-mediated transgene silencing, all of which point to the significance of the piRNA pathways’ epigenetic functions [156]. In K562 cells, a CML line cell line, overexpression of Hiwi by lentivirus dramatically reduced cell growth and resulted in apparent apoptosis. Hiwi-expressing K562 cells produced tumors in BALB/c nude mice that were smaller than those produced by the control group [157]. In the U937 cell line, overexpressing piRNA 011186 accelerated the cell cycle, reduced apoptosis, and inhibited CDKN2B gene expression [80]. AML may now have new prognostic and diagnostic biomarkers such as piRNA 32877 and piRNA 33195 [158]. In multiple myeloma patients and cell lines, piRNA-823 was elevated and positively linked with the clinical stage. By controlling DNA methylation and angiogenesis in multiple myeloma, piRNA-823 aids in carcinogenesis [81].

## 3. Non-Coding RNAs as Biomarkers and Therapeutic Targets in Leukemia

Structured non-coding RNAs (ncRNAs) are crucial components of several biological processes, including protein synthesis, signaling, RNA processing, and gene expression control. This class of RNA, which includes rRNA, tRNA, snRNA, and snoRNA, primarily serves as the structural framework of higher complexes such as ribosomes. Even though they have structural roles, some also play regulatory duties. Several studies have involved these RNAs in several disorders, including cancer [159]. However, more information is needed concerning their function in leukemia; therefore, we attempt to outline what is known.

It has been established that rRNA mutations have a significant role in cancers. Additionally, CML shows the same behavior. It has been demonstrated that mutations, by decreasing the translational efficiency of early CLL cells, rewire the translation program of the ribosomal protein S15 (RPS15), a component of the 40S ribosomal subunit [160]. In conjunction with amplifying the MLL and CBFA2 genes, rRNA gene amplification has also been seen in case reports. Interestingly, in AML with an NPM1 mutation, the long non-coding RNA HOXB-AS3, which controls AML cell proliferation in vitro and in vivo, also regulates ribosomal RNA transcription [79].

In malignancies, dysregulation of transfer RNA (tRNA) encourages the translation of tumorigenic mRNAs [161]. Enhanced mitochondrial bioenergetics are caused by altered valine tRNA biosynthesis in T-ALL. A crucial T-ALL oncogene, NOTCH1, transcriptionally regulates valine aminoacyl tRNA synthetase expression, highlighting the function of oncogenic transcriptional programs in balancing tRNA supply and demand [162]. Ts-43 and Ts-44 are down-regulated 3 to 5-fold in CLL compared to B cells and are generated from different pre-tRNAHis genes. Additionally, the expression of phenylalanine tRNA (tRNAPhe) differs between normal and cancerous cells [163].

Spliceosome’s primary component is snRNA. In total, 40–80% of individuals with myelodysplastic syndrome (MDS), notably those with ringed sideroblasts, have spliceosome gene alterations [164]. In two recent large whole-genome sequencing investigations, 10% of patients with CLL have mutations in the spliceosome gene SF3B1 [165,166]. It has been shown that CLL cases have a worse prognosis due to the mutation in U1 snRNA [167]. Numerous tumors have also been found to often harbor missense mutations in the genes SRSF2, SF3B1, ZRSR2, and U2AF1, particularly in specific subtypes of leukemia [168]. The Jab1/MPN domain and other regions of the PRPF8 gene are dispersed across the length of the gene, and the missense mutations were most commonly found in primary and secondary acute myeloid leukemia [169]. These splicing factor mutations cause RNA splicing patterns to change, including splicing dysregulation, incorrect splicing of genes involved in tumorigenesis, and the encouragement of tumor-genic isoforms [170].

Small nucleolar RNAs (snoRNAs) are non-coding RNAs that range in length from 60 to 300 nucleotides. Based on their structure and function, they may be divided into two groups: C/D box snoRNAs, which are crucial for modifying ribosomal RNA via 2’-O-methylation, and H/ACA box snoRNAs, which are involved in the pseudouridylation of rRNA. AML1-ETO improved C/D box snoRNP and rRNA 2’-O-methylation in AML with chromatin (8;21) translocation and facilitated the interaction between amino-terminal enhancers and RNA helicase DDX21, which ultimately sped up cell proliferation in vitro and in vivo [171,172]. Compared to CD34+ progenitors, monocytes, and granulocytes, primary AML patient samples consistently had greater levels of SNORD42A expression. In several human leukemia cell lines, the C/D box containing SNORD42A was discovered to be a crucial modulator for the survival and proliferation of AML cells [82]. Patients with improved clinical outcomes for CLL were identified by having decreased expression of SNORA74 and SNORD116-18 [173]. It has been discovered that IGHV mutation in CLL cases could regulate the expression of a group of 20 snoRNAs related to survival rate. Overexpression of snoRNA host gene SNHG12 was associated with a poor prognosis and carcinogenesis in diffuse large B-cell lymphoma via sponging miR-195 [83]. Despite the fact that there are many examples of snoRNAs’ participation in leukemia that cannot be addressed here because of the vast quantity of data, these RNAs’ modes of action are varied. Generally speaking, these RNAs might indicate their function by sponging miRNAs, accumulating pre-45S ribosomal units, or enhancing oncogene translation to support the cancer state [174].

## 4. Non-Coding RNAs Role in Medicine and Their Limitation

Since ncRNAs play various roles in cells and their dysregulation has been linked to several illnesses, including malignancies, they may make excellent molecular therapeutic targets. This has led to the development of several RNA-based treatments, such as antisense oligonucleotides, small interfering RNAs, short hairpin RNAs, ASO anti-miRs, miRNA mimics, miRNA sponges, therapeutic circular RNAs, and CRISPR-Cas9-based gene editing [9,175]. These medications regulate RNAs in various ways, such as by focusing on pre-mRNA splicing. In addition, other RNA treatments, including more recent ones such as miRNA mimics and antimiRs, are in phase II or III clinical development; nevertheless, no lncRNA-based medicines have reached the clinic [176]. Unfortunately, most research on ncRNAs and associated therapeutics do not focus on hematologic disorders rather than solid tumors as their primary target for therapy. While several antisense oligonucleotides have been developed, including Fomivirsen (for CMV infection) [177], Mipomersen, Golodirsen, and Viltolarsen (for Duchenne muscular dystrophy) [178], and Inclisiran (for familial hypercholesterolemia) [179], nothing has been conducted to treat leukemia. Prexigebersen (BP1001-A), an antisense nucleotide, is noteworthy as a rare example. It is now being tested in phase II clinical study for the treatment of patients with AML in conjunction with chemotherapy [180]. Despite the small number of projects in this field, there is a restriction in the laboratory sector [181]. Many of these components contain chemical modifications and sequences that could cause immunogenicity, and our immune system can recognize both single-stranded (ss) and double-stranded (ds) RNA via a variety of extra- and intracellular PAMP receptors, raising the possibility that immunogenicity of this type of treatment could be a problem [182].

Additionally, because these components depend on their nucleotide sequences, specificity may be a limiting factor because the potency of an RNA therapeutic’s on-target specificity and the absence of off-target and undesirable on-target effects are both indicators of quality [183]. In addition, the delivery method could be another field to investigate. Efficient delivery of oligonucleotides is challenging owing to their instability, negative charge, and hydrophilic nature of RNAs preventing diffusion through cell membranes. Overcoming this section could eliminate many obstacles; however, even in this era, a state of art delivery method is needed [184]. Regardless of therapeutic opportunities and limitations, candidating the best ncRNA and finding the best option is also a significant problem. Before spreading the NGS techniques, which could efficiently show the differential expression of ncRNAs, qRT-PCR was the primary source of identification. The study’s power is constrained by some unfavorable characteristics for both methodologies.

It should be noted that some ncRNAs, including miRNAs, require specialized cDNA synthesis kits [185]. Additionally, many ncRNAs, particularly lncRNA, create RNA structures that may obstruct accurate RT-PCR measurement [186]. The use of standard qRT-PCR is limited by low expression, low copy number, and—more importantly—the presence of ncRNA clusters, which prevents us from utilizing relative quantification or cyber green and forces us to employ absolute quantification using probes [187]. The proper investigation of the internal control and standardization choices is another obstacle to ncRNA quantification [188]. Additionally, even though several sequencing techniques exist for quantifying ncRNAs, the program and process are not as established as those for mRNA quantification, which has been the focus of well-known and sophisticated bioinformatics programs [189,190].

## 5. Future Directions

The significant role of ncRNAs in biological networks and recent advances in molecular cancer have made them favorable therapeutic targeting agents. Inhibition of these elements can help suppress more critical mechanisms involved in tumorigenesis. The therapeutic sensitivity to conventional chemotherapy can be restored by suppressing the expression of ncRNAs, and drug resistance can be overcome. While RNA expression could always be a diagnostic and prognostic marker, ncRNAs as a potential candidate with their presence in serum and blood plasma is a favorable choice. Investigating ncRNA’s functions in cancer may be essential, given their significant roles in cells. Despite extensive research on ncRNAs, there are still several challenges to overcome. Notable issues include the process of cell transformation, carrier safety, differential targeting of malignant and healthy cells, and, more crucially, the immunological questions surrounding targeting agents (siRNAs, miRNAs, Crisper-based treatment, etc.). However, by overcoming these obstacles, targeting ncRNAs may soon rank among the most widely used treatments for various malignancies such as leukemia.

## Figures and Tables

**Figure 1 genes-14-00131-f001:**
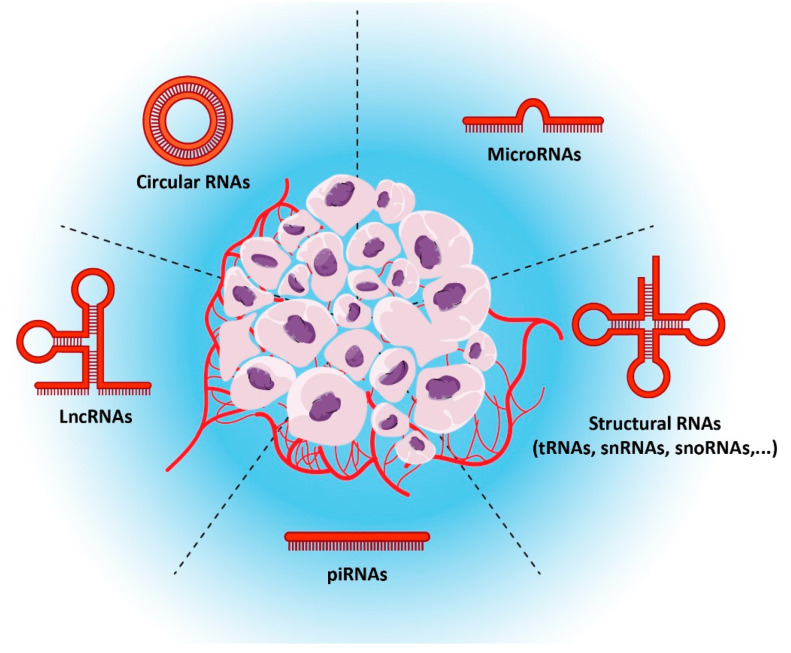
Different types of non-coding RNA.

**Figure 2 genes-14-00131-f002:**
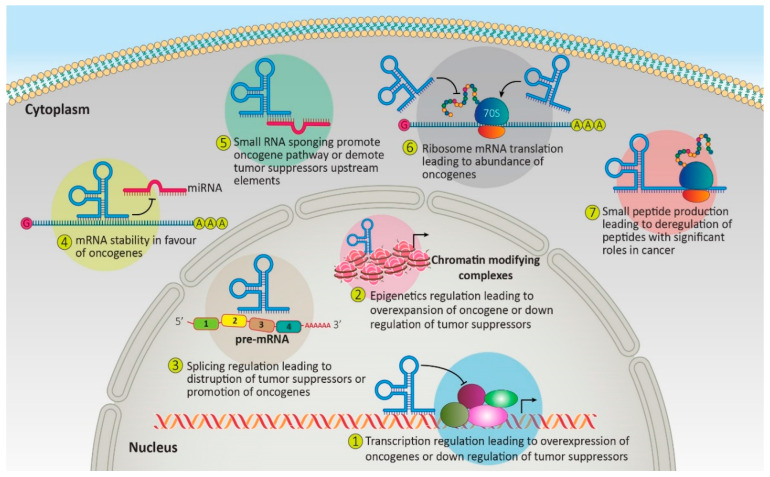
The proposed mechanism of action for non-coding RNAs to regulate oncogenes and tumor suppressor’s network.

**Table 1 genes-14-00131-t001:** Summary of the proposed mechanism of non-coding RNAs in leukemia.

Type	Effector	Target	Reference(s)
miRNA	miR-194-5p	BCLAF1	[21]
	miR-628	IGF-1R	[22]
	miR-143	Autophagy	[23]
	miR-126	SHIP1 and CEBPB	[24,25]
	miR-133	Evi1	[26]
	miR-223	G1/S cell cycle	[27,28,29]
	miR-181a	Cell proliferation and enhanced cell cycling	[30]
	miR-128	Apoptosis, genome instability, and drug sensitivity	[31]
	miR-125b	Cell survival and lowered apoptosis	[32]
	miR-222	ETS1	[33]
	miR-223	FOXO1	[34]
	miR-142-3p	Drug resistance	[35]
	miR-26b	PIK3CD	[36]
	miR-26	Ikaros	[37]
	miR-30a	NOTCH1 and NOTCH2	[38]
	miR-221	STAT5	[39]
	miR-497	CDKN2A/B	[40]
	miR-328	MALAT1	[41]
	miR-214	ABCB1	[42]
	miR-10a	USF1	[43]
	miR-203	Proliferation	[44]
	miR-29b	Abl-1	[45]
	miR-320	ABL	[46]
	miR -181a and miR-181b	Drug resistance	[47,48]
	miR-125b	GRK2 and PUMA	[49]
lncRNA	RP11-137H2.4	NR3C1	[50]
	CRNDE	Proliferation	[51]
	PVT1	Myc	[52]
	TUG1	Proliferation	[53]
	CCAT1	Proliferation and myeloid cell differentiation	[54]
	NR-104098	EZH2	[55]
	LINC00265	PI3K-Akt	[56]
	NEAT1	Proliferation and drug resistance	[57]
	HOTAIR	MRP1	[58,59]
	SGNH5	Drug resistance	[60]
	Hmrhl	Tumor development	[61]
circRNA	Circ-DLEU2	proliferation	[62]
	Circ_0004277	miR-134-5p and SSBP2	[63]
	CircSPI1	SPI1	[64]
	CircMYBL2	FLT3	[65,66,67]
	Circ _004136	miR-142 and miR-29a	[68]
	CircRNF13	miR-1224-5p	[69]
	Circ_0121582	miR-224	[70]
	Circ_100290	miR-203	[71]
	Circ_0079480	miR-655-3p	[72]
	Circ_0002483	Cell proliferation and cell cycle arrest	[73]
	CircBA1	Cell growth and medication resistance	[74]
	CircCRKL	miR-877-5p	[75]
	Circ-0058493	miR-548b-3p	[76]
	Circ-0132266	miR-337-3p	[77]
	CircZNF91	miR-1283	[78]
	HOXB-AS3	Ribosomal RNA transcription	[79]
piRNA and snoRNA	11186	CDKN2B	[80]
	823	DNA methylation and angiogenesis	[81]
	SNORD42A	Proliferation	[82]
	SNHG12	miR-195	[83]

## Data Availability

Not applicable.

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
