# Peer review of "The Promising Role of Non-Coding RNAs as Biomarkers and Therapeutic Targets for Leukemia"

_genes, 2023, doi:10.3390/genes14010131_

Round 1

Reviewer 1 Report

In the submitted manuscript "The promising role of non-coding RNAs as biomarkers and 3 therapeutic agents in leukemia: a complete guide" the authors gave a literature review of so-far knowledge on ncRNAs in leukemia and their potential application in diagnosis and therapy. Although the authors presented a comprehensive review of results related to the subject, in order to make it more focused and potentially attractive to the readers, I suggest the following significant changes:

-please introduce the table with a summary of all findings presented in the text. In this way, you can highlight the most promising ncRNAs as diagnostic or therapeutic targets. 

-please design a figure that includes the roles of ncRNAs in the pathophysiology of leukemia

-considering that the focus of the paper should be on diagnostics and therapy, an additional chapter should be added regarding current challenges in the utilization of ncRNAs in diagnosis (for example, preanalytical and analytical challenges in quantification of ncRNAs, absolute vs. relative quantification, standardization, and harmonization) and therapy (specificity of the targets, delivery of the drugs, etc.)

-the section named discussion is redundant. Please change into either future perspectives or conclusion.

Author Response

-please introduce the table with a summary of all findings presented in the text. In this way, you can highlight the most promising ncRNAs as diagnostic or therapeutic targets. 

Thank you for your time and input of yours which enlighten us and our manuscript. We have added a table and summarized the most important ncRNA whose mechanism has been elucidated to avoid unnecessary data.

-please design a figure that includes the roles of ncRNAs in the pathophysiology of leukemia

Thanks for your thoughtful comment. We have added a figure showing the mechanism of ncRNA in which they could influence pathophysiology of leukemia

-considering that the focus of the paper should be on diagnostics and therapy, an additional chapter should be added regarding current challenges in the utilization of ncRNAs in diagnosis (for example, preanalytical and analytical challenges in quantification of ncRNAs, absolute vs. relative quantification, standardization, and harmonization) and therapy (specificity of the targets, delivery of the drugs, etc.)

We have added a section as your instruction. Many thanks.

-the section named discussion is redundant. Please change into either future perspectives or conclusions.

Thank you for your warm help. It has been changed.

Reviewer 2 Report

In this review, the authors collect data in the literature about the emerging role of ncRNAs and their involvement in the processes of oncogenesis, with a particular focus on acute leukemias.

While I found some merit in their work, several issues of concern are present

11)     The paper begins with an introduction a basic reference about oncohematologic diseases and their signs and symptoms. This part, although not the focus, should be improved. In particular, Line 34-40 are inadequate in explaining how leukemias develop and its manifestations. Besides, in Line 34 please correct the term chronic lymphoblastic leukemia (lymphocytic)

22)      I found the long list of about 8 pages of ncRNAs with short sentences characterized by a succession of “name-function” very difficult to read.

Eg: "Let-7a, miR-99b, miR-146a, miR-150, miR-155, miR-191, and miR-1246 were among the miRNAs Hornick et al. identified as being elevated in AML exosomes from the serum of model mice. These serum exosomal miRNAs might be used to detect AML early. MiR-26a-5p and miR-101-3p, were markedly elevated in AML, whereas miR-23b-5p, miR-120 339-3p, and miR-425-5p were markedly reduced in exosomes [38]."

I think this part should be modified, and summarizing Tables  should be added.

33)     I agree with the authors that ncRNAs represent a challenge to be approached in the future for a target treatment of hematologic diseases. Most of the studies conducted on ncRNAs and related therapies unfortunately do not concern hematologic diseases but find main application in treatment of solid tumors.

I think the clinical advances should be mentioned in the review, with particular reference to antisense oligonucleotides.

Approved drugs are few, mainly for liver and neuromuscular diseases: ( eg Fomivirsen (antisense oligonucleotide used for CMV infection); Mipomersen, Golodirsen and Viltolarsen (for treatment of Duchenne muscular dystrophy); Mipomersen, Inclisiran (used in familial hypercholesterolemia).

Besides, clinical trials in leukemia could also be mentioned ( eg, Prexigebersen (BP1001-A)  an antisense nucleotide currently in phase II clinical trial for the treatment of patients with AML in combination with chemotherapy).

44)     Please check the references, ad several errors are present

- References 27 and 81 are the same

- References 38 and 39 are the same

- References 62 and 63 are the same

- References 70 and 106 are the same

 5) English is acceptable, but several mistakes and typos are present (eg- Line 34: wide spread -----widespread or common , check Line 40 ,  Lline 60 available?.....)

Author Response

  • The paper begins with an introduction a basic reference about oncohematologic diseases and their signs and symptoms. This part, although not the focus, should be improved. In particular, Line 34-40 are inadequate in explaining how leukemias develop and its manifestations. Besides, in Line 34 please correct the term chronic lymphoblastic leukemia (lymphocytic)

Thank you for your most precious comments which we learn and elevate the manuscript based on them. We have altered this section with deeper details.

2)      I found the long list of about 8 pages of ncRNAs with short sentences characterized by a succession of “name-function” very difficult to read. Eg: "Let-7a, miR-99b, miR-146a, miR-150, miR-155, miR-191, and miR-1246 were among the miRNAs Hornick et al. identified as being elevated in AML exosomes from the serum of model mice. These serum exosomal miRNAs might be used to detect AML early. MiR-26a-5p and miR-101-3p, were markedly elevated in AML, whereas miR-23b-5p, miR-120 339-3p, and miR-425-5p were markedly reduced in exosomes [38]."

 We have changed the grammar in this section and rewritten some parts as your instruction. As this manuscript wants to introduce a whole view of ncRNAs role in leukemia we could not find a better way to show ncRNA main role as briefly as possible, It already has around 200 references and more detail need more specialized review articles. Unfortunately, this is a limitation that we could not overcome on this occasion.

I think this part should be modified, and summarizing Tables  should be added.

Thank you for your thoughtful comments. A table with key information on ncRNAs has been added.

  • I agree with the authors that ncRNAs represent a challenge to be approached in the future for a target treatment of hematologic diseases. Most of the studies conducted on ncRNAs and related therapies unfortunately do not concern hematologic diseases but find main application in treatment of solid tumors. I think the clinical advances should be mentioned in the review, with particular reference to antisense oligonucleotides. Approved drugs are few, mainly for liver and neuromuscular diseases: ( eg Fomivirsen (antisense oligonucleotide used for CMV infection); Mipomersen, Golodirsen and Viltolarsen (for treatment of Duchenne muscular dystrophy); Mipomersen, Inclisiran (used in familial hypercholesterolemia). Besides, clinical trials in leukemia could also be mentioned ( eg, Prexigebersen (BP1001-A)  an antisense nucleotide currently in phase II clinical trial for the treatment of patients with AML in combination with chemotherapy).

Thank you for your comments. A section named as Non-coding RNAs role in medicine and their limitation has been added. As you mentioned the studies are few and we add the examples you mentioned which cover the subject almost entirely.

4)     Please check the references, ad several errors are present

- References 27 and 81 are the same

- References 38 and 39 are the same

- References 62 and 63 are the same

- References 70 and 106 are the same

Thank you and sorry for the inconvenience we have corrected these mistakes.

 5) English is acceptable, but several mistakes and typos are present (eg- Line 34: wide spread -----widespread or common , check Line 40 ,  Lline 60 available?.....)

Thank you for your accurate comment. We have corrected the mistake as we could.

Round 2

Reviewer 1 Report

The authors have made substantial changes in the manuscript. I would just propose small additions to figure 2. In the current form, the figure addresses the roles of ncRNAs more broadly. It would be helpful to specify particular mechanisms implicated in the pathophysiology of leukemia.

Author Response

Thank you for your help which enhances this manuscript.

We have added the potential role of ncRNA in pathogenesis via the promotion of oncogene and demotion of tumor suppressors. While this subject is too broad we tried to summarize this in the figure as short as possible and hope this lives up to your expectation.

Warmly

Reviewer 2 Report

The authors adequately adressed my concerns

Author Response

Thank you.